# *HLA-DRB1*07:01* and **08:02* Alleles Confer a Protective Effect Against ACPA-Positive Rheumatoid Arthritis in a Latin American Admixed Population

**DOI:** 10.3390/biology9120467

**Published:** 2020-12-14

**Authors:** Patricia Castro-Santos, Jordi Olloquequi, Ricardo A. Verdugo, Miguel A. Gutiérrez, Carmen Pinochet, Luis A. Quiñones, Roberto Díaz-Peña

**Affiliations:** 1Facultad de Ciencias de la Salud, Universidad Autónoma de Chile, Talca 3460000, Chile; patricassan@gmail.com (P.C.-S.); jordiog82@gmail.com (J.O.); 2Inmunología, Centro de Investigaciones Biomédicas (CINBIO), Universidad de Vigo, 36310 Vigo, Spain; 3Programa de Genética Humana, ICBM, Facultad de Medicina, Universidad de Chile, Santiago 8389100, Chile; raverdugo@uchile.cl; 4Departamento de Oncología Básico-Clínico, Facultad de Medicina, Universidad de Chile, Santiago 8389100, Chile; 5Rheumatology, Almirante Nef Naval Hospital, Viña del Mar, Valparaíso 2340000, Chile; magutierrezt2004@gmail.com; 6School of Medicine, Valparaíso University, Valparaíso 2340000, Chile; 7Hospital Regional de Talca, Talca 3460000, Chile; cmpinochet@gmail.com; 8Laboratory of Chemical Carcinogenesis and Pharmacogenetics, Department of Basic-Clinical Oncology, Faculty of Medicine, University of Chile, Santiago 8320000, Chile; 9Latin American Network for Implementation and Validation of Clinical Pharmacogenomics Guidelines (RELIVAF-CYTED), 28015 Madrid, Spain

**Keywords:** HLA, HLA-DRB1, rheumatoid arthritis, ACPA, Latin American, admixed population

## Abstract

**Simple Summary:**

The focus of the genetic studies on rheumatoid arthritis has been on Europeans or European-derived populations, with limited information available on other populations, especially in Latin America. The aim of the present study was to test previously reported HLA markers, the most important genetic contributors for the risk of developing rheumatoid arthritis, associated with anti-citrullinated protein antibodies-positive rheumatoid arthritis in a Latin American population with admixed ancestry. Our study shows for the first time an association between *HLA-DRB1*07:01* and **08:02* alleles and protection against anti-citrullinated protein antibodies-positive rheumatoid arthritis in the Chilean population. In addition, our results seem to support the existence of differentiated genomic patterns in Chilean, and probably other Latin American populations, that are not the same that the found in Europeans regarding to loci repeatedly involved in rheumatoid arthritis. Identifying relationships between *HLA-DRB1* alleles and rheumatoid arthritis is important for identifying disease associations in different ethnic groups in order to reach a better understanding of rheumatoid arthritis worldwide.

**Abstract:**

*HLA-DRB1* shared epitope (SE) alleles are important genetic contributors for the risk of developing anti-citrullinated protein antibodies (ACPA)-positive rheumatoid arthritis (RA), particularly in Caucasians. We aimed to analyze the contribution of HLA-DRB1 alleles and single nucleotide polymorphisms (SNPs) within the major histocompatibility complex (MHC) region to the susceptibility to develop ACPA-positive RA in a Latin American (LA) population with admixed ancestry. A total of 289 ACPA-positive RA patients and 510 controls were enrolled in this study. The presence of *HLA-DRB1*04:01*, **09:01* and **10:01* was increased in ACPA-positive RA patients compared with healthy controls (*p* < 0.0001, *p* < 0.001 and *p* < 0.01, respectively), whereas *DRB1*07:01* and **08:02* was associated with a decreased risk of ACPA-positive RA (*p* < 0.001 and *p* < 0.01, respectively). These results showed a strong correlation with estimates from studies in Asians but not in Caucasian populations. The present study describes the protective effects of the *HLA-DRB1*07:01* and **08:02* alleles in ACPA-positive RA patients in a LA population for the first time. Identifying relationships between *HLA-DRB1* alleles and RA is important for identifying disease associations in different ethnic groups in order to reach a better understanding of RA worldwide.

## 1. Introduction

Rheumatoid arthritis (RA) is a multifactorial, progressive, systemic and inflammatory autoimmune disease that affects approximately 1% of the population worldwide [1]. The prevalence, clinical manifestation and prognosis of RA varies across populations and is influenced by ethnic, socioeconomic, and geographic differences [1]. Most of the studies have been carried out in countries from the North America and North Europe, finding prevalence of 0.5–1.1% [1]. On the other hand, the lowest prevalence values have been found in parts of Asia and Africa, and the highest in Native American populations [1]. Few data are available for developing countries. In Latin America, several studies have been developed in limited regions to a country. This might not be representative of the nation’s prevalence, considering ethnic and cultural differences [2], and genetic variability [3,4], that exist in each country this area of the world. Estimates show a RA prevalence ranging from 0.3 to 2% [5,6,7,8,9,10,11,12].

Since it was first described in 1978 [13], genetic susceptibility to RA has been repeatedly linked to chromosome 6, and specifically to the human “major histocompatibility complex, MHC” (HLA) region. The genes within this region code for paramount proteins involved in the immune defense against invading pathogens. Specifically, MHC class II molecules are present on the membrane of antigen presenting cells (APC). Through these molecules, these APCs display antigens derived from extracellular and membrane proteins, leading to the activation of CD4+ T cells responses. MHC class II proteins are expressed from three gene regions (DR, DQ and DP). Particularly, the *HLA-DRB1* shared epitope (SE) has been shown to encode an amino acid sequence largely associated with RA susceptibility and progression (^70^QRRAA^74^, ^70^QKRAA^74^, or ^70^RRRAA^74^) [14]. The existence of this SE suggests that HLA molecule could bind to molecules similar to bacterial antigens or the named “arthritogenic self-peptides”, hence shaping may drive the T-cell-antigen repertoire to an autoimmune response. Notwithstanding, this arthritogenic peptide remains to be identified.

Anti-citrullinated protein autoantibodies (ACPAs) show high specificity for RA and have become a substantial component of the current ACR-European League Against Rheumatism (EULAR) classification criteria for RA [15]. It has been hypothesized that apoptosis and/or necrosis of pulmonary cells could originate an increased citrullination of proteins in the lungs, due to the increased activation of peptidyl-arginine deiminases enzymes (PAD). The binding of these citrullinated proteins to HLA-DR molecules on APC and their presentation to T cells, followed by B cells activation, may trigger the development of high titters of ACPA. Events so diverse as infection, trauma, exercise, etc., could lead to citrullination of proteins in the joints and the subsequent formation of immune complexes between the citrullinated proteins and ACPA, followed by the binding of immune complexes to the Fc receptors of the synovial macrophages and triggering a chronic inflammatory response.

*HLA-DRB1* gene confers a high polymorphism to the HLA-DR complex [16]. Since Exon 2 of *HLA-DRB1*, which encodes the antigen recognition site, is the most variable region, differences in antigen presentation can be largely related to polymorphisms in *HLA-DRB1*. Interestingly, ethnic differences in the distribution of specific *HLA-DRB1* SE alleles worldwide have been reported [17], and these alleles have been shown to be the most important genetic risk factors to develop ACPA-positive RA in Caucasians [18]. In turn, in studies in Latin American (LA) populations with a large proportion of European ancestry some *HLA-DRB1* alleles have also been associated with RA [18], and also a meta-analysis seemed to revalidate the SE hypothesis in LA populations [19,20] In spite of this, whether the association of *HLA-DRB1* and RA is valid in all ethnic groups remains unclear. In this sense, genome-wide association studies (GWAS) have identified an increasing number of single-nucleotide polymorphisms (SNPs) rendering potential susceptibility to RA in Caucasian and/or Asian populations [21,22]. However, none of these studies GWASs were performed on LA ethnicities [23]. Consequently, the aim of the present study was to analyze the contribution of *HLA-DRB1* alleles and SNPs within the MHC region to the susceptibility to develop ACPA-positive RA in the Chilean population, a LA population characterized by a high admixed ancestry.

## 2. Patients and Methods

### 2.1. Study Population

A total of 289 RA patients and 510 healthy controls were enrolled in this study. RA patients were consecutive recruited between January 2015 and December 2017 at the Rheumatology Unit of Hospital de Talca, Talca, Chile and at the Rheumatology Unit of the Health Network, the Pontifical Catholic University of Chile, Santiago de Chile, Chile. All RA patients were ACPA-positive and diagnosed following the 2010 American College of Rheumatology/European League Against Rheumatism (ACR/EULAR) classification criteria [15]. Patient data were retrospectively analyzed from their anonymised medical records. The control population consisted of 510 matched unrelated healthy blood donors whose samples were collected between January 2015 and May 2020 at Casa del Donante, Talca, Chile and at the CQF Laboratory of the University of Chile (Special Control DNA Biobank), Santiago de Chile, Chile. The study was approved by the Ethical Committee of the “Servicio de Salud del Maule”, Chile (registration Nº04/2014); and all individuals gave their written informed consent prior to enrolling in the study.

### 2.2. Characterization of HLA-DRB1 Alleles and Levels of ACPA

*HLA-DRB1* genotyping was performed using an HLA-SSO typing kit (Tepnel Lifecodes Corporation, Stamford, CT, USA) according to the manufacturer’s instructions. ACPA status was determined by using enzyme-linked immunosorbent assay (ELISA) according to the manufacturer’s instructions (Euro-Diagnostica AB, Malmö, Sweden). The cut-off value for positivity was set at 25 IU/mL.

### 2.3. SNP Genotyping

Twenty-four SNPs were chosen for genotyping to check whether previously reported SNP markers within the HLA region (SNPs showing *p* < 0.05, after Bonferroni correction) were associated with RA in LA populations with admixed ancestry [20]. These SNPs were genotyped in our LA admixed population using the OpenArray^®^™ *TaqMan* platform (Applied Biosystems Inc., Waltham, MA, USA) at the Centro Nacional de Genotipado at the Santiago de Compostela node, Spain: rs2027856, rs2157335, rs2239802, rs2395178, rs2395182, rs3104389, rs2858332, rs3129768, rs3129867, rs3129882, rs3129886, rs3129888, rs3135335, rs34102154, rs3998158, rs4959028, rs7775228, rs9268614, rs9268844, rs9271640, rs9275224, rs9275580, rs9275582 and rs9501626.

### 2.4. Statistical Analysis

The distribution of *HLA-DRB1* alleles was compared between RA patients and controls using Fisher’s exact test and two-by-two contingency tables with or without each allele. We used the SPSS v.22 statistical software to estimate the odds ratios (OR) and 95% confidence intervals (95% CI). For comparisons of the *HLA-DRB1* allele frequencies, Bonferroni correction was performed by multiplying the *p* value by the number of *DRB1* alleles tested (*n* = 36) to give the corrected *p*-value. The following quality criteria were used for the SNP genotyping data: minor allele frequency (MAF) <0.01, Hardy–Weinberg equilibrium (HWE) *p* < 0.001, and/or missingness >0.1. Allele frequencies were compared between RA patients and controls by Chi-Square analysis, and the OR and 95% CI were calculated using the PLINK software [24].

## 3. Results

Table 1 displays the characteristics for the 289 RA patients who were enrolled in the present study. The mean age was 48 years and 84.8% of the patients were women, thus confirming the high prevalence of RA in LA women reported in other studies [25]. The mean duration of the disease was 8 years. There were no differences in the sociodemographic parameters between the RA patient and control groups.

To investigate the influence of *HLA-DRB1* SE alleles on the risk of developing ACPA-positive RA, we genotyped *HLA-DRB1* alleles in 289 RA patients and 510 healthy controls. The frequencies of the *HLA-DRB1* alleles in the Chilean controls were slightly different from previously reported results [26,27,28] (Table 2), which was probably due to the diverse origin of the samples from a population with admixed ancestry, most of which is of Mapuche and European origin [4]. *HLA-DRB1* SE allele positivity was significantly associated with ACPA-positive RA in the Chilean population [61.8% vs. 33.1%, *p* < 10^−12^, OR = 2.28 (2.43–4.44)] (Table 3). Furthermore, the presence of *DRB1*04:01*, **09:01* and **10:01* was increased in ACPA-positive RA patients compared with healthy controls [*p* < 0.0001, <0.001, and <0.01 and OR = 3.5 (2.05–5.97), 3.46 (2.01–5.96), and 4.11 (1.98–8.52), respectively], whereas the presence of *DRB1*07:01* and **08:02* was associated with a decreased risk of ACPA-positive RA [*p* < 001, and <0.01 and 0.30 (0.18–0.52), and 0.09 (0.02–0.38), respectively].

Figure 1 shows the single marker allelic association results of the studied SNPs. Of the 24 SNPs included, four could not be tested because the assay failed in the genotyping phase. (SNPs rs3104389, rs34102154, rs9271640 and rs9275224). The observed genotypic SNP distributions were consistent with Hardy–Weinberg equilibrium (HWE) expectations in both controls and RA patients (*p* > 0.05). Seven markers exhibited significant association (*p* ≤ 0.05) with RA after Bonferroni correction (rs3135335, rs3129867, rs2395178, rs9268844, rs9275224, rs9275580, rs3998158; Appendix A). The associations of the remaining genes included in the study were not significant. Single-SNP association values in Chile showed similar profiles with estimates from in LA populations with admixed ancestry [20] (Figure 1).

We also compared the odds ratios of the *HLA-DRB1* alleles in our data with reported allelic odds ratios for *HLA-DRB1* alleles in a large study of ACPA-positive RA [29] (Figure 2). Specifically, this study included 5018 RA cases and 14974 controls from the UK, Sweden, Canada, and the United States, as well as a South Korean collection of 616 RA cases and 675 controls, all of them being positive for ACPA. There was little correlation between data belonging to Caucasian populations and our data (r  =  0.41, *p*  =  0.02). The effect sizes for each of the *HLA-DRB1* alleles were similar with the reported results, except most of the *HLA-DRB1* alleles associated with risk of ACPA-positive RA in our study (*DRB1*04:01*, **07:01*, **08:01*, **09:01* and **10:01*). On the contrary, our data showed a strong correlation with those from the Korean population (r  =  0.63, *p*  =  10^−4^). In addition, the effect size of the *DRB1*08:02* allele, whose presence was associated with a decreased risk of ACPA-positive RA (Table 3), was different when compared to the Korean population.

## 4. Discussion

Many studies have established associations between the susceptibility to ACPA-positive RA and some *HLA-DRB1* alleles [18], mainly *DRB1*01*, **04* and **10*, in Caucasian and Asian populations. However, these studies have not been replicated in populations with a large admixture of Amerindian ancestry. In this study, we analyzed the contribution of *HLA-DRB1* alleles to the susceptibility to ACPA-positive RA in the Chilean population.

Our results showed an association between certain DRB1 alleles and RA susceptibility in the Chilean population. Thus, the frequencies of *DRB1*04:01*, **09:01* and **10:01* were increased in ACPA-positive RA patients, whereas *DRB1*07:01* and **08:02* demonstrated a lower frequency in ACPA-positive RA patients. These results are in line with those reporting genetic associations of RA with *HLA-DRB1* SE alleles in LA populations [18]. Moreover, our data also suggest an association between the that ACPA-positive RA and specific HLA polymorphisms from Caucasian and Asian populations. In this sense, *DRB1*09:01* confers a strong susceptibility to RA in Asian populations [30]; however, it differs from other SE alleles at position 74, while sharing a common sequence of three Arg as the *DRB1*10:01* susceptibility alleles found in Caucasians. In addition, *DRB1*09:01* was associated with the presence of ACPA in individuals with Native American as well as Mexican American ancestry [31]. Surprisingly, the frequency of the *HLA-DRB1* alleles in our control population was not similar to previous reports from the Chilean population [26,27,28]. This could be explained by the fact that the distribution of genetic structure of Chileans is not the same throughout the country [4]. Finally, ORs in Chileans showed little correlation with estimations based in studies made in Europeans, but a strong association with Asians. Since RA is associated with loci involved in immune responses, it is also highly associated with local adaptations and disease resistance. Populations have their own historical particularity, and the exposure to diverse disease factors or traits differs from one to others. Hence, our results seems to support the existence of differentiated genomic patterns in Chilean, and probably other LA populations, that are not the same that the found in Europeans regarding to loci repeatedly involved in RA [32].

The common amino acid sequence at positions 70–74 in the DRβ chain showed an increased RA risk (QKRAA, QRRAA or RRRAA), while alleles that changed from Q/R => D at position 70 exerted a protective effect against RA, especially those alleles containing the ^70^DERAA^74^ sequence, such as *DRB1*01:03*, **13:01*, **13:02* or **04:02*. We also observed a trend in the protective role of *HLA-DRB1*04:02* in RA susceptibility [31] in the Chilean population. This is relevant, since *DRB1*04:02* confers a susceptibility to pemphigus vulgaris (PV) [33], a relatively rare autoimmune disease that is potentially lethal and characterized by blistering of the skin and mucosal membranes. While it is important to define the association of this allele across populations worldwide, the mechanistic basis for the dual role of *HLA-DRB1*04:02* in the HLA-disease association, with a protective role in RA and as a genetic risk factor for PV, garners great interest and is not currently understood.

Our study shows for the first time an association between *HLA-DRB1*07:01* and **08:02* alleles and protection against ACPA-positive RA in the Chilean population. In 2002, de Vries N et al. reported significant protective effects for *DRB1*07* in RA in a study with 167 RA patients and 166 healthy controls, all of whom were Caucasian [34]. More recently, *DRB1*07* has been associated with protection against RA in populations of North Africa [31]. Regarding *HLA-DRB1*08*, this allele was positively associated with systemic lupus erythematosus (SLE) [35] and systemic sclerosis (SSc) [36]. *HLA-DRB1*08:02* allele was found associated with bucillamine-induced proteinuria in Japanese RA patients [37]. Similar to methotrexate, d-penicillamine, sulfasalazine, gold salts and anti-malarial drugs, bucillamine is a disease-modifying anti-rheumatic drug (DMARD) frequently used in RA treatment. Further research is needed to define specific the role of this allele in RA.

We analyzed SNPs previously associated with RA in LA populations with admixed ancestry [20] in our test population from Chile. In the previous study by López Herráez et al., 196.524 SNP were genotyped in 1.475 RA patients and 1.213 controls from different LA countries, including 135 patients and 78 controls from a Chilean population. Our results are quite consistent to these previous ones, with 35% of SNPs showing a significant association, thus corroborating the complex relationship between HLA and RA in LA populations. On the other hand, we acknowledge that the non-homogeneous origin of the samples and the different size of the sample among populations limit the implications of the results. In this sense, however, single-SNP association values in Chile showed similar profiles compared to LA populations with admixed ancestry described in this study. When the genetic association was examined only in ACPA-positive RA patients [20], the involvement of the HLA region became more significant, particularly for SNP rs9275224 within DQB1/DQA2. Our data revealed the most significant association for the same SNP. GWAS allow us to collect data of single SNP and subsequently infer HLA alleles by HLA imputation in recent studies [38,39,40]. However, this probably is not recommended in LA populations due to the insufficient information available and the complexity of the admixture. Recently, a GWAS was performed on individuals from four countries in Latin America who were diagnosed as having SLE [41]. HLA allele imputation was performed using the HIBAG program with its corresponding Hispanic reference data set [40]. Efforts to clarify the role of HLA in RA and the differences in LA populations should be done. So far, classic HLA typing keeps being the best tool.

## 5. Conclusions

The present study describes for the first time a protective effect of the *HLA-DRB1*07:01* and **08:01* alleles in ACPA-positive RA patients of a LA population. It also describes the association of *HLA-DRB1* SE alleles commonly found ACPA-positive RA from Asian/Caucasian populations in our LA cohort. Finally, this research seems to support the existence of different genomic patterns in LA populations than those found in Europeans regarding to loci repeatedly associated with RA. These results stress the importance of analyzing the relationships between *HLA-DRB1* alleles and RA risk in different ethnic groups to contribute to a better understanding of RA worldwide.

## Figures and Tables

**Figure 1 biology-09-00467-f001:**
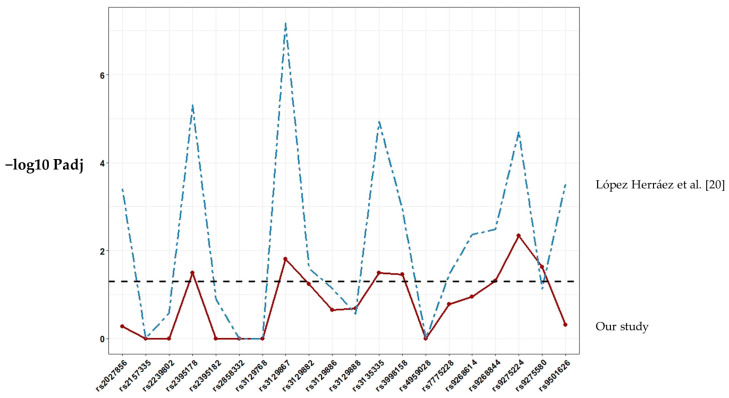
Association results for SNP markers within the HLA region included in this study. The score of the −log10 Padj values for each SNP association is shown as a continuous line plot. Abbreviations: Padj, *p*-value using Bonferroni correction.

**Figure 2 biology-09-00467-f002:**
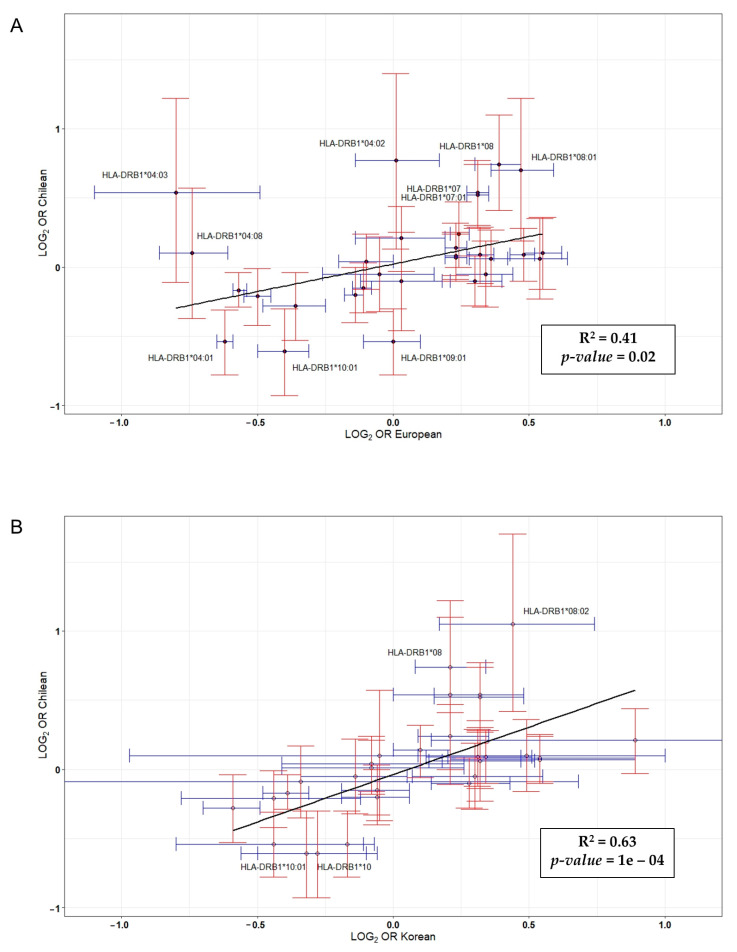
Correlation between log(odds ratio) from data published in a genome-wide association (GWA) study carried out in Caucasian (**A**) and Korean (**B**) populations versus log(odds ratio) reported in this study [29].

**Table 1 biology-09-00467-t001:** Characteristics of the patients with rheumatoid arthritis.

Demographic Characteristics	RA Patients	Healthy Controls
Women, *n* (%)	245 (84.8)	399 (78.2)
Men, *n* (%)	44 (15.2)	111 (21.8)
Age, years (median, range)	48 (20–75)	42 (25–72)
**Clinical Parameters**		
Disease duration, years (median, range)	8 (1–42)	-
Age of onset, years (median, range)	40 (15–67)	-

**Table 2 biology-09-00467-t002:** Comparison between *HLA-DRB1* SE alleles frequencies in previous reports and in our controls from Chile.

HLA-DRB1 SE Alleles	Frequency			
	Our cohort (*n* = 510)	Schäfer C et al. [26]	Rey D et al. [27]	Thorsby E et al. [28]
01:01	0.08	0.02	0.01	0.00
01:02	0.03	0.03	0.00	0.24
04:01	0.04	0.01	0.00	0.05
04:04	0.08	0.01	0.01	0.02
04:05	0.05	0.02	0.00	0.02
04:08	0.02	0.01	0.00	0.00
10:01	0.02	0.01	0.01	0.00
14:02	0.06	0.04	0.15	0.00

**Table 3 biology-09-00467-t003:** Distribution of *HLA–DRB1* alleles in patients with rheumatoid arthritis and healthy controls.

HLA-DRB1 Allele	RA Cases (*n* = 289)		Controls (*n* = 510)		*p*-Value	BONF	OR (CI 95%)
	*n*	%	*n*	%			
**SE-positive†**	179	61.76	169	33.10	2.83 × 10^−15^	1.02 × 10^−13^	2.28 (2.43–4.44)
**01:01**	36	12.35	42	8.16	NS	NS	-
**01:02**	10	3.53	14	2.82	NS	NS	-
**01:03**	0	0.00	4	0.73	NS	NS	-
**03:01**	43	14.71	87	16.97	NS	NS	-
**03:05**	0	0.00	2	0.36	NS	NS	-
**04:01**	41	14.12	23	4.44	2.18 × 10^−6^	7.85 × 10^−5^	3.5 (2.05–5.97)
**04:02**	2	0.59	20	3.83	0.01	NS	0.17 (0.04–0.74)
**04:03**	2	0.59	12	2.37	NS	NS	-
**04:04**	37	12.94	42	8.25	NS	NS	-
**04:05**	26	8.82	25	4.97	NS	NS	-
**04:07**	19	6.47	52	10.27	NS	NS	-
**04:08**	5	1.76	11	2.18	NS	NS	-
**04:10**	0	0.00	1	0.20	NS	NS	-
**07:01**	17	5.88	88	17.33	7.08 × 10^−6^	2.55 × 10^−4^	0.30 (0.18–0.52)
**07:11**	0	0.00	2	0.36	NS	NS	-
**08:01**	3	1.18	26	5.14	5 × 10^−3^	NS	0.20 (0.06–0.65)
**08:02**	2	0.59	36	7.05	4.2 × 10^−5^	1.51 × 10^−3^	0.09 (0.02–0.38)
**08:06**	0	0.00	1	0.20	NS	NS	-
**08:11**	2	0.59	0	0.00	NS	NS	-
**09:01**	39	13.53	22	4.28	4.44 × 10^−6^	1.60 × 10^−4^	3.46 (2.01–5.96)
**10:01**	24	8.24	11	2.07	7.15 × 10^−5^	2.57 × 10^−3^	4.11 (1.98–8.52)
**11:01**	29	10.00	58	11.47	NS	NS	-
**11:04**	0	0.00	3	0.59	NS	NS	-
**12:01**	0	0.00	6	1.12	NS	NS	-
**13:01**	17	5.88	37	7.27	NS	NS	-
**13:02**	14	4.71	28	5.50	NS	NS	-
**13:03**	0	0.00	5	0.98	NS	NS	-
**13:05**	2	0.59	0	0.00	NS	NS	-
**14:01**	22	7.65	35	6.85	NS	NS	-
**14:02**	20	7.06	29	5.62	NS	NS	-
**14:13**	2	0.59	0	0.00	NS	NS	-
**15:01**	19	6.47	55	10.77	NS	NS	-
**15:02**	17	5.88	27	5.28	NS	NS	-
**15:03**	0	0.00	1	0.20	NS	NS	-
**16:01**	0	0.00	3	0.59	NS	NS	-
**16:02**	32	11.18	58	11.29	NS	NS	-

*HLA-DRB1* shared epitope alleles, including **01:01*, **01:02*, **04:01*, **04:04*, **04:05*, **04:08*, **10:01* and **14:02* alleles. Abbreviations: NS, not significant; RA, rheumatoid arthritis; SE, shared epitope.

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
