# Peer review of "HLA-DRB1*07:01 and *08:02 Alleles Confer a Protective Effect Against ACPA-Positive Rheumatoid Arthritis in a Latin American Admixed Population"

_biology, 2020, doi:10.3390/biology9120467_

Round 1

Reviewer 1 Report

The authors report results from a genetic study for HLA-DRB1 alleles and SNPs within the MHC region to the susceptibility todevelopment of ACPA-positive RA in the Latin American Chilean population. Few studies have been conducted in admixted Latin American populations, thus this study is relevant to the genetics of RA etiology.

The results include the novel finding of a protective effect of the HLA-DRB1*07:01 and *08:01 alleles in ACPA+ Latin American RA patients.

This study was well-performed, and in this Reviewer's opinion, should be published.

Author Response

  1. This study was well-performed, and in this Reviewer's opinion, should be published.

R: We thank the reviewer comment.

Reviewer 2 Report

The authors study the HLA association with ACPA positive RA in Chile, as well as separate SNPs within the MHC region, which were described as significant in a previous study on LA populations. This study verifies the association with SE alleles, and additionally describe a protective association with DRB1*07:01 and *08:02 alleles.

  1. There are several nomenclature issues, which need to be addressed for clarity and consistent with the published literature. When the authors refer to a gene or allele, it needs to italicized. Also, the alleles do not bind, but the HLA molecule (line 103).
  2. The introduction needs to be restructured and shortened. Many sentences need references (for example line 115), and need to clarify that these are examples of theories stated for the initiation and propagation of the autoimmune response in RA. Figure 1 is not necessary and does not add to the text. APCs do not activate B cells and needs to be corrected.
  3. In the methods, please clarify where the cohort was selected, time frame, location, and how the controls were chosen (demographics, location, etc). It is mentioned later on that the controls were matched by sociodemographic parameters, but these are not explained anywhere. Also, how was ACPA positivity defined?
  4. In the methods, please clarify which SNPs were chosen and for what reason.
  5. Table 1 needs units for age, disease duration, and can add the appropriate numbers for the control population as well.
  6. Regarding table 2, were the differences statistically significant?
  7. In table 3, please define SE (which alleles were included)
  8. In the discussion, it is not quite clear why PV was mentioned. The same HLA molecule can present PV epitopes.

Author Response

  1. There are several nomenclature issues, which need to be addressed for clarity and consistent with the published literature. When the authors refer to a gene or allele, it needs to italicized. Also, the alleles do not bind, but the HLA molecule (line 103).

R: Thank you for the comment. We have now corrected all matters of nomenclature. We have also amended the phrase in question. Please see revised manuscript.

  1. The introduction needs to be restructured and shortened. Many sentences need references (for example line 115), and need to clarify that these are examples of theories stated for the initiation and propagation of the autoimmune response in RA. Figure 1 is not necessary and does not add to the text. APCs do not activate B cells and needs to be corrected.

R: Thank you for the comment. According to the Reviewer’s suggestion, we have restructured and shortened the introduction. We have also deleted the Figure 1. Please see revised manuscript. Regarding the comment “APCs do not activate B cells and needs to be corrected”, it was a mistake., Our apologies for this. We have rewritten the sentence: “The binding of these citrullinated proteins to HLA-DR molecules on APC and their presentation to T cells, followed by B cells activation, may trigger the development of high titters of ACPA” (page 2, lines 88-90).

  1. In the methods, please clarify where the cohort was selected, time frame, location, and how the controls were chosen (demographics, location, etc). It is mentioned later on that the controls were matched by sociodemographic parameters, but these are not explained anywhere. Also, how was ACPA positivity defined?

R: We have included a brief description about the cohort selected (RA patients and healthy controls) in the Patients and Methods section (2.1. Study population, page 3). We have also added the demographic characteristics of the controls in the table 1. Likewise, we have included how ACPA positivity was defined (2.2. Characterization of HLA-DRB1 alleles and levels of ACPA, page 3).

  1. In the methods, please clarify which SNPs were chosen and for what reason.

R: According to the reviewer, we have included a more detailed explanation about the selection of the SNPs, at the same time as that we indicate the specific SNPs (2.3. SNP genotyping, page 3)

  1. Table 1 needs units for age, disease duration, and can add the appropriate numbers for the control population as well.

R: Thank you for the comment. We have included all this information in the table 1.

  1. Regarding table 2, were the differences statistically significant?

R: As to the Reviewer’s query, we wish to point out that Table 2 was not intended to reflect statistical differences. Instead, we wanted to show the lack of homogeneity between our results and those from three published reports, in order to highlight the importance of having homogeneous populations in genetic studies like these, since it brings the opportunity to unveil different genetic patterns and their association with RA.

  1. In table 3, please define SE (which alleles were included)

R: Done (please see revised manuscript, page 5, lines 172-173)

  1. In the discussion, it is not quite clear why PV was mentioned. The same HLA molecule can present PV epitopes.

R: Thank you for the comment. We just feel like it is a curious thing concerning the HLA-DRB1*04:02 allele, and that it would need to be clarified: the mechanistic basis for their dual role, with a protective role in rheumatoid arthritis and as a genetic risk factor for pemphigus vulgaris. If the reviewer does not consider it appropriate, this part can be deleted.
